# Development of Korea Airport Pavement Condition Index for Panel Rating

Nam-Hyun Cho [1] , Hong-Joon Kwon [2], Young-Chan Suh [3] and Jangrak Kim [4],*

1 Airport Research Institute, Incheon International Airport Corporation, Jung-gu, Incheon 22382, Korea
2 Department of Highway & Transportation Research, Korea Institute of Civil Engineering and Building Technology, Goyang-si 10223, Korea
3 Department of Transportation & Logistics Engineering, College of Engineering Sciences, ERICA Campus, Hanyang University, Ansan-si 15588, Korea
4 SOC Business Department, ROADTECH, Ansan-si 15588, Korea
* Correspondence: jaksal93@hanmail.net; Tel.: +82-10-2843-2127

**Abstract:** Airports strive to prevent safety issues, such as foreign object debris (FOD), by pavement management using the pavement condition index (PCI). The index is used in decision-making processes for overall pavement maintenance and repair, such as the prevention of additional damage due to cracks and the like. However, considering the current situation in Korea where mostly mid-sized and large commercial airports exist, problems regarding direct applications of the existing PCI deduct value have been consistently pointed out. In addition, as the relationship between the PCI and whether maintenance and repair are required is unrealistic, there have been difficulties in communication between maintenance and repair staff and decision makers. Therefore, to resolve these problems, this study first analyzed the calculation procedure of the existing PCI and then redefined the main distress type of Korean airport pavements. In addition, a deduct value curve (DVC) in terms of the severity level for six main distress factors of asphalt pavements and eight main distress factors of concrete pavements and a corrected deduct value curve (CDVC) for multiple distresses in terms of the pavement form were developed using panel rating, which is an engineering approach, by forming an airport pavement expert panel. Finally, a Korea airport pavement condition index (KPCI) was proposed using the curves, and the field application results were compared against the existing PCI to examine the adequacy of the KPCI. As a result, the developed criteria showed an overall trend lower than existing PCI. Moreover, it was verified that this trend increases with worsening pavement condition. It appears that a more discriminating evaluation may be possible when determining pavement conditions by PCI results of the developed criteria.

**Keywords:** pavement condition index; Korea airport pavement condition index; panel rating; deduct value; corrected deduct value

## 1. Introduction

The pavement condition index (PCI) is a quantitative index for evaluating pavement conditions which has been used in airport pavement management systems (APMS) since the early 1980s [1]; numerous airports in Korea and other countries have evaluated pavements using this index. The index has provided sound engineering data for pavement conditions and has particularly been used for preventing potential problems, such as foreign object debris (FOD), such as shown in Figure 1, and in decision-making processes for overall pavement maintenance and repair such as the prevention of additional damage due to cracks and the like. In particular, FOD may be an element of debris due to distress on the pavement surface, it is very important to maintain an appropriate pavement condition through evaluation. Ensuring proper maintenance of airport pavement and technical parameters is one of the key factors for ensuring operational safety in the ground maneuvering sector in terms of aircraft–airport surface contact [2].

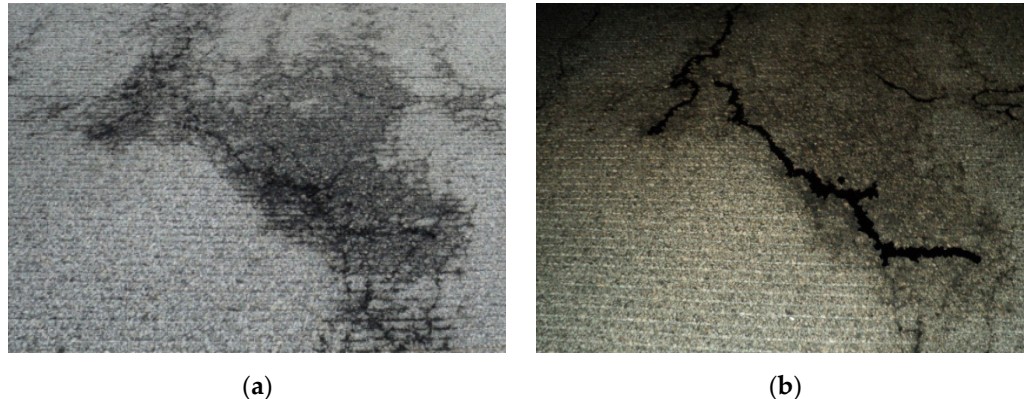

(**a**)                                     (**b**)

**Figure 1.** Example showing potential of FOD due to pavement distress and repair; (**a**) High FOD potential, (**b**) low FOD potential.

However, Broten and Sombre [3] pointed out potential problems and limitations when it comes to the applicability of the PCI calculation procedure in accordance with the differences between the environment at the time of development and the current environment. In particular, it is pointed out that PCI decreases after maintenance. Sun et al. [4] classified a total of 13 distress types by combining the patching size classification and spalling location classification among the 15 distress types classified in ASTM D 5340. And a new PCI calculation method and a corrected deduct value curve (CDVC) for multiple distresses was developed, though the deduct value curve (DVC) was used without modifications. Jackson [5] considered the limitation of being unable to properly consider current pavement distresses in adjusting the distress types used for PCI evaluation or when placing more weights on main distresses.

Various studies have been conducted to overcome these limitations. First, the Virginia Department of Transportation (VDOT) developed its own new pavement condition indicators based on the existing PCI [6]. An Italian study conducted a similar procedure [7]. They added two types of distress and developed a new density/deductive curve to apply it to their needs and pavement surface condition. In addition, in India [8], road pavement evaluation methods based on the Overall Pavement Status Index (OPCI) have been developed. The OPCI model includes four indicators: pavement distress index, surface roughness index, pavement structural capacity index, and skid resistance.

In addition, it has been found that the ratings of the existing PCI developed for airports of various scales in the United States showed significant differences considering the management levels of Korean airports, which mainly consist of mid-sized and large commercial airports [9]. As a result of compared and examined deduct values (DV) in terms of the distress to verify that the PCI decreases after maintenance and repair if block cracks of a low severity level with a distress density of approximately 25% or higher are patched. Moreover, they pointed out the unreasonable problem where the pavement condition was satisfactory even though maintenance and repair were required. This is believed to be because of the communication problems between the maintenance and repair staff and decision makers [10]. This study was conducted to resolve these problems and to develop an airport PCI suitable for Korean circumstances. Accordingly, we analyzed the existing PCI calculation procedure, redefined the main distresses of Korean airport pavements by forming an airport pavement expert panel, and newly developed a DVC and a CDVC using the panel rating, which is an engineer's experience approach. Moreover, the field application results were compared and examined against the existing PCI to ensure the adequacy of the Korea airport pavement condition index (KPCI), which is calculated using the curves.

## 2. Materials and Methods

### 2.1. Background

Assuming a condition without distresses as a maximum of 100 [11,12], the PCI is calculated by subtracting the deduct value (DV), which is a function of the distress type, severity, and density on the pavement surface using Equation (1). In case of multiple distresses, a corrected value is calculated [13,14].

$$PCI = C - \sum_{i=1}^{p} \sum_{i=1}^{m_i} a\left(T_i,\, S_j,\, D_{ij}\right) F(t, d) \tag{1}$$

Here, PCI is the pavement condition index, C is a constant depending on the desired maximum scale value (100), and $a\left(T_i, S_j, D_{ij}\right)$ is the deduct weighing value depending on the distress type ($T_i$), level of severity ($S_j$), and density of distress ($D_{ij}$). i, j, and p are the counter for the distress types (i), counter for the severity levels (j), and total number of distress types for the pavement type under consideration (p), respectively. The number of severity levels for the i th type of distress are represented by $m_i$, and $F(t, d)$ is an adjustment factor for multiple distresses that vary with the total summed deduct value (t) and number of deducts (d).

The PCI calculation is carried out in sample units. The size of an asphalt pavement is defined to be $464.5 \pm 185.8$ m², and a concrete pavement is assumed to consist of $20 \pm 8$ slabs [14]. If a given sample unit contains different distress types or different severity levels but of the same type, they are treated as multiple distresses and an adjustment factor is applied. A corrected deduct value (CDV) calculation process using the total deduct value (TDV) and the number (q) of distresses to be considered is then carried out [15,16]. The TDV is determined by adding all deduct values from each distress condition observed. And CDV is determined based on the TDV and the number of distress conditions observed with individual deduct values over five points. The PCI is calculated as in Equation (2).

$$PCI = 100 - CDV \tag{2}$$

This PCI calculation enables automatic road analysis, such as ARAN systems [17]. The system is based on the data on the pavement deterioration obtained during the measurement. The data is collected in the form of photos of the pavement made with cameras and a three-dimensional image of the pavement obtained as a result of three-dimensional scanning [18]. Unfortunately, the automatically obtained data are then processed manually by humans, which still affects the quality of information about the actual technical condition of the assessed pavement. ARAN was also used to collect distress data to find potential relations with pavement roughness and pavement condition [19].

To overcome the limitations of these analytical techniques, Iranian scientists in their study [20] presented attempts to develop an alternative method for determining the PCI using optimization techniques based on artificial neural networks and a genetic algorithm. The proposed approach may help in the future to reliably extrapolate the PCI index.

### 2.2. Development Process of KPCI

The development process, shown in Figure 2, was derived in detail by analyzing the definitions of the distress types in terms of the pavement form, severity classifications, deduct values of individual distresses, and CDV calculation procedures for multiple distresses.

The most difficult part in the PCI development process is to determine the DV and CDV [21]. Ideally, the deduct value must be based on the effects of the distress situation (type, severity, and density) of a pavement on the structural integrity and operating conditions of the pavement. However, measuring these effects will require considerable research efforts for comprehensive field evaluation and analytical or theoretical determination. However, a reasonable DV that is suitable in all regions can be derived using a subjective approach based on the collective evaluation of skilled pavement technicians [22].

Therefore, a panel of 12 members including an experienced professor and experts on the field of pavement was first formed. Subsequently, a DV was derived by panel rating.

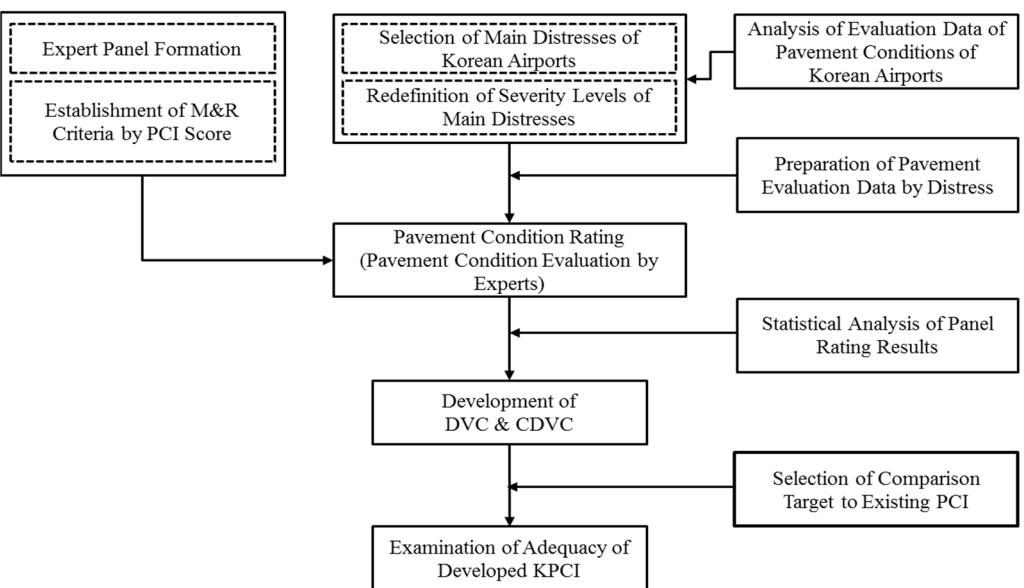

**Figure 2.** Development Process of KPCI.

### 2.3. Pavement Condition Index Rating Scale

An evaluation criterion classified into separate subjective categories, as listed in Table 1, is required to determine the definitions of the distress and DV. This criterion provides a PCI range and M&R standard using pavement condition rating, which are technical indices required for reasonable and subjective evaluation of the effect of a given distress. The PCI range becomes a standard for determining the pavement condition, and the M&R type is determined accordingly [23]. One of the limitations of the existing PCI is the lack of practical connection between the evaluation results and the need for repair. Therefore, a standard for the pavement condition evaluation was prepared in advance to ensure a practical connection between the PCI and the need for repair. This paper prepared the M&R standard in terms of the PCI range by incorporating the circumstances in Korea and results of expert group meetings based on the PCI rating scales suggested in ASTM D5340, as listed in Table 1.

**Table 1.** PCI Score Rating and M&R Type.

| PCI Range | Pavement Condition | Maintenance/Rehabilitation (M&R) Type | |
|---|---|---|---|
| | | **Asphalt Pavement** | **Concrete Pavement** |
| 90–100 | Good | Routine maintenance<br>- Crack sealing | Routine maintenance<br>- Crack sealing<br>Joint replacement |
| 80–89 | Fair | Routine maintenance<br>- Crack sealing<br>- Patching | Routine maintenance<br>- Crack sealing<br>- Joint replacement<br>Partial section fix |
| 60–79 | Caution | Preventive maintenance<br>- Crack sealing<br>- Patching<br>Partial Grind/Overlay | Preventive maintenance<br>- Crack sealing<br>- Joint replacement<br>Partial section fix |
| 40–59 | Poor | Overlay and Grind/Overlay<br>(7.5~10 cm)<br>Construction | Asphalt overlay or<br>Grind/Overlay (over 10 cm)<br>Construction |
| 0–39 | Very Poor | Construction | Construction |

## 2.4. Definitions of Distress Type and Severity Level

The distress types existing in pavements must be identified and classified to develop a pavement condition index [13]. Therefore, the APMS data of 15 Korean airports affected by different environmental and traffic conditions from the past ten years were collected and analyzed. Based on these data and experiences of pavement experts, the main distresses were selected.

As a result, six types (alligator cracking, block cracking, joint reflection cracking, longitudinal and transverse cracking, patching and utility cut patching, and rutting) of distresses were considered the main distresses. In the case of concrete pavements, spalling was not classified by location (joint and corner); existing small and large and utility cut patching was combined into patching, and shrinkage cracking that did not have a separate severity level criterion was treated with scaling, map cracking, and crazing. Thus, the similar distresses among the 16 types were combined to yield 13 different types of distresses, among which eight types (ASR, corner cracking, cracks, durability cracking, intersecting cracks, patching, scaling, and spalling) were classified as distresses. The targets to be developed are selected, as listed in Table 2, and ASTM D 5340 was applied to the other distresses. High (H), medium (M), and low (L) of ASTM D5340 were applied to the distress severity level.

**Table 2.** Distress Type of Developed Deduct Value Curve.

| Asphalt Pavement | | Concrete Pavement | |
|---|---|---|---|
| **Distress Type of Deduct Value Curve Developed** | **Distress Type of Existing Deduct Value Curve Used** | **Distress Type of Deduct Value Curve Developed** | **Distress Type of Existing Deduct Value Curve Used** |
| Alligator Cracking Block Cracking Joint Reflection Cracking Longitudinal and Transverse Cracking Patching and Utility Cut Patching Rutting | Bleeding Corrugation Depression Jet-Blast Erosion Oil Spillage Polished Aggregate Raveling Shoving Slippage Cracking Swell Weathering | Alkali Silica Reaction (ASR) Corner Cracking Cracks; Longitudinal, Transverse and Diagonal Durability ("D") Cracking Intersecting Cracks Patching, Small, Large and Utility Cuts Scaling, Map Cracking, Shrinkage Cracking Spalling (Longitudinal and Transverse Joint, Corner) | Blowup Joint Seal Damage Popouts Pumping Settlement or Faulting |

## 2.5. Panel Rating

Practical photograph data of each distress type were collected from the PMS data of the Korean airports. A conclusion was reached that sample units comprising an asphalt pavement of size 450 m$^2$ and a concrete pavement comprising 18 slabs would provide adequate pavement areas for PCI evaluation, and the calculation was performed by conducting a preliminary analysis on the collected data with the help of expert discussions.

The sample unit was sufficiently wide for meaningful distress measurement. The distress density was determined by dividing the surface area of the particular distress measured for a particular severity level by the total area of the sample unit in case of the asphalt pavement. As the linear cracking (longitudinal or transverse) is measured in terms of length (m), the distress area was calculated considering an affected width of 0.3 m to determine the density. The distress density of the concrete pavement was determined by dividing the number of slabs with a particular distress of a particular severity level by the total number of slabs of the sample unit. The evaluation was carried out for five distress density levels. The distress values of each asphalt pavement were evaluated for levels in the ranges of 0.1–1.5%, 1.6–4.9%, 5–14.9%, 15–39.9%, and 40–75%, and the distress values of each concrete pavement were evaluated for density levels of 1/18, 2/18, 4/18, 8/18, and

12/18. This is based on the fact that sections with a distress density of 80% do not exist in Korean airports.

The panel independently and subjectively carried out the evaluation in accordance with the rating criteria, as listed in Table 1. The pavement condition rating (PCR) evaluated by the panel was calculated for each sample unit and for each density level [24]. Considering the deviations within the panel, the PCR was calculated as moving average ($\overline{\overline{PCR}}$) excluding the bottom and top 10 percentiles. Here, moving average (MA) is a traditional statistical method for analyzing the trend of time series data fluctuations, which can be used to predict and smooth data [25]. The MA model removes the anomaly data in the manner of averaging neighboring samples [26]. The average DV was calculated using Equation (3).

$$DV = 100 - \overline{PCR} \qquad (3)$$

where DV is the average DV, the value 100 is a constant depending on the desired maximum scale value, and $\overline{PCR}$ is the average PCR excluding the top and bottom 10 percentiles.

Figure 3a shows the PCR with respect to the distress density of severity "H" for concrete pavement cracks (longitudinal, transverse, and diagonal), the average ($\overline{\overline{PCR}}$) excluding the top and bottom 10 percentiles, and the Figure 3b shows the DVC, which is a regression curve passing through the average DV.

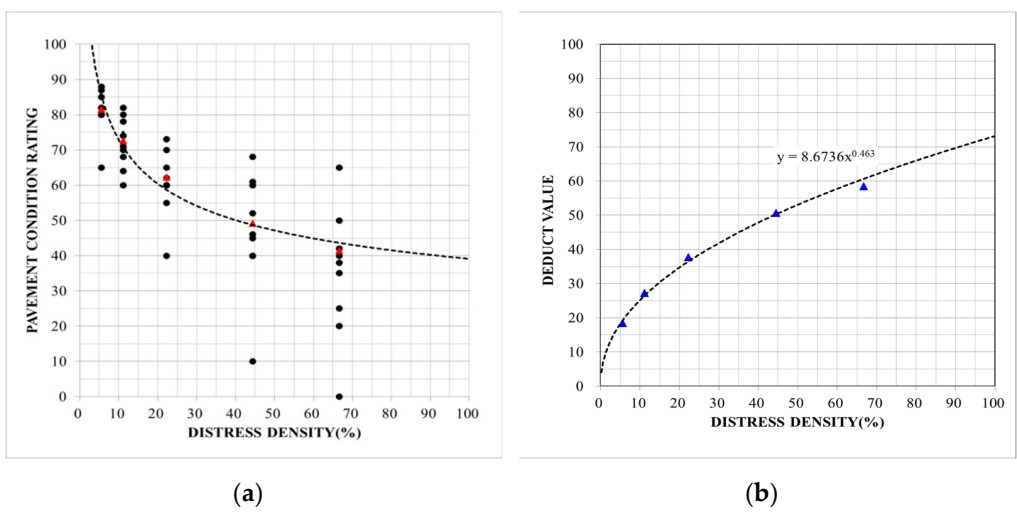

(**a**)　　　　　　　　　　　(**b**)

**Figure 3.** Example showing PCR, DV, and DVC for linear cracking in concrete pavement; (**a**) PCR and average PCR, (**b**) average deduct value and deduct value curve.

In case of two or more distress types or severity levels in a pavement, the DVs cannot simply be added, and the TDV must be adjusted by incorporating the number of deducts (or the distress types and the levels of severity) and the magnitude of the sum of deduct values. Such adjustments for multiple distresses were determined by evaluating pavement sections including two to five distress types or severity levels. The CDV was calculated using the sum of the DVs calculated using individual DVCs and a corrected value obtained by subtracting the PCR calculated by the panel rating from 100 for each sample unit.

### 3. Results

#### 3.1. Deduct Value Curve and Corrected Deduct Value Curve

The numbers of DVCs developed for the three severity levels for each distress type of six asphalt pavements and eight concrete pavements were 18 and 24, respectively. These values reflected smooth curve forms passing through each point in the distress density–average DV plots. Moreover, modeling for the CDVs including two to five distress types and severity levels was carried out. The DVs lower than 5 were not considered, as they hardly affect the pavement conditions.

Figures 4 and 5 show the distress density with respect to the DV of the asphalt pavements for six types of distresses, and of the concrete pavements for eight types of distresses. For comparison purposes, the DVCs of ASTM D 5340 are displayed using dotted lines. The standard deviation of the DV by the panel rating increases with the increases in the severity level and distress density.

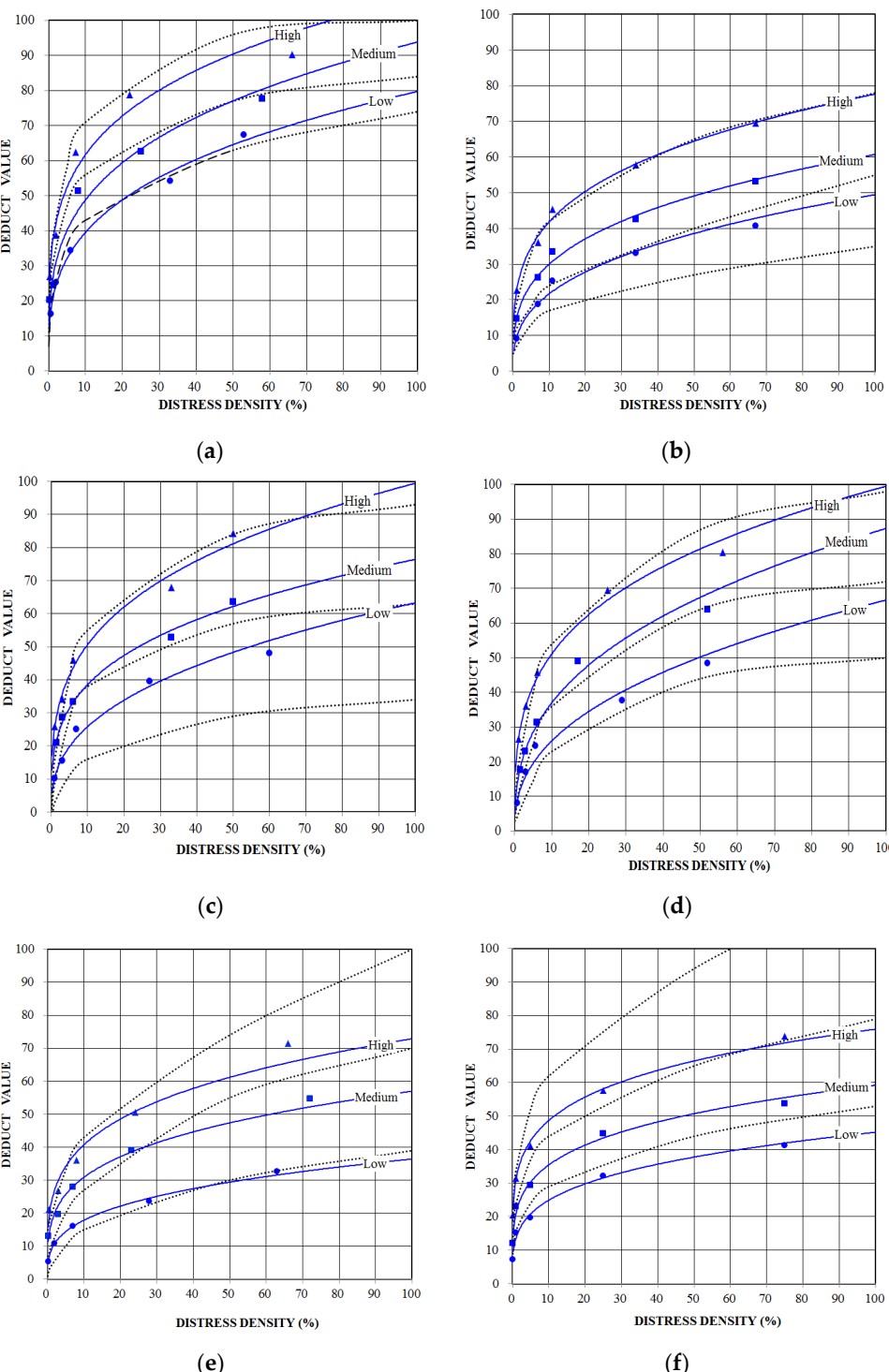

**Figure 4.** Comparisons between developed deduct value curves and ASTM D 5360 in asphalt pavement; (**a**) Alligator cracking, (**b**) block cracking, (**c**) reflection cracking, (**d**) longitudinal and transverse cracking, (**e**) patching and utility cut patching, (**f**) rutting.

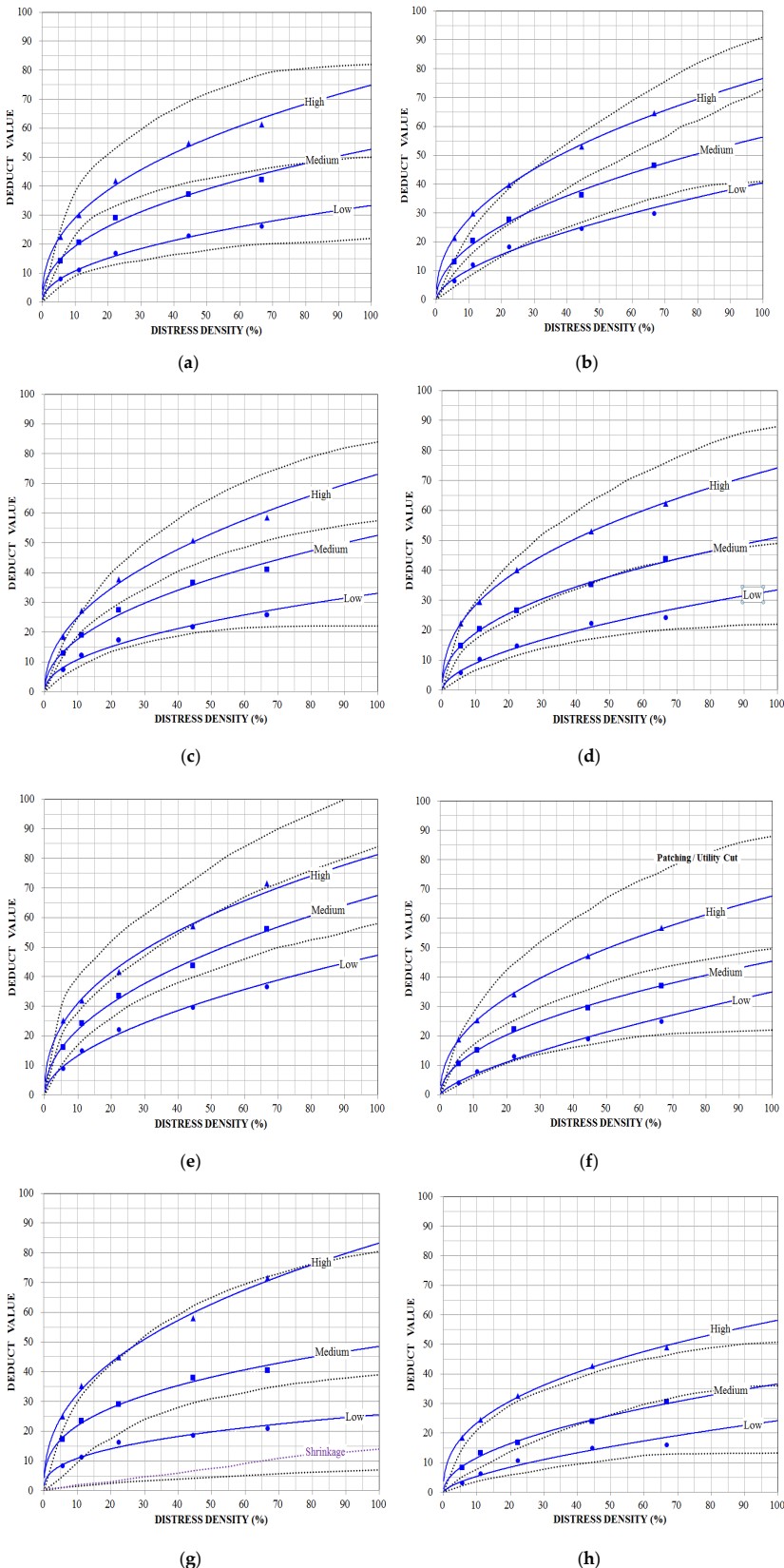

**Figure 5.** Comparisons between developed deduct value curves and ASTM D 5360 in concrete pavement; (**a**) Alkali Silica Reaction (ASR), (**b**) corner cracking, (**c**) cracks; longitudinal, transverse, and diagonal, (**d**) durability ("D") cracking, (**e**) intersecting cracks, (**f**) patching, small, large, and utility cuts, (**g**) scaling, map cracking, shrinkage cracking, (**h**) spalling (longitudinal and transverse joint, and corner).

In general, the shapes of the curves are important as they indicate relative effects of the distress density on the pavement conditions. The DV showed an abruptly increasing trend up to approximately 15% of the distress density; however, this trend was less evident thereafter. Comparing the developed DVC and the DVC of ASTM D 5340, they showed largely low or similar values in the "H" and "M" severities and largely high values in the "L" severity. The higher the density, the higher was the DV of the developed KPCI in all the three severity levels; however, the DV of the developed KPCI became lower than that given in ASTM D 5340 as it increased. The differences in the DV between the severity levels were lower than that given in ASTM D 5340. This is because of incorporating the Korean airport pavement management level that takes into consideration the importance of occurrences of distresses even at low severity, and initial distress management.

Figures 6 and 7 show the effects of the different distress types of the asphalt pavements and concrete pavements on the DV. The alligator cracking of the asphalt pavements and the intersecting cracks of the concrete pavements resulted in the highest deduct value. This is due to incorporating the recognition of the panel members participating in the panel rating for FOD possibilities when it comes to the alligator cracking and severity of the intersecting cracks.

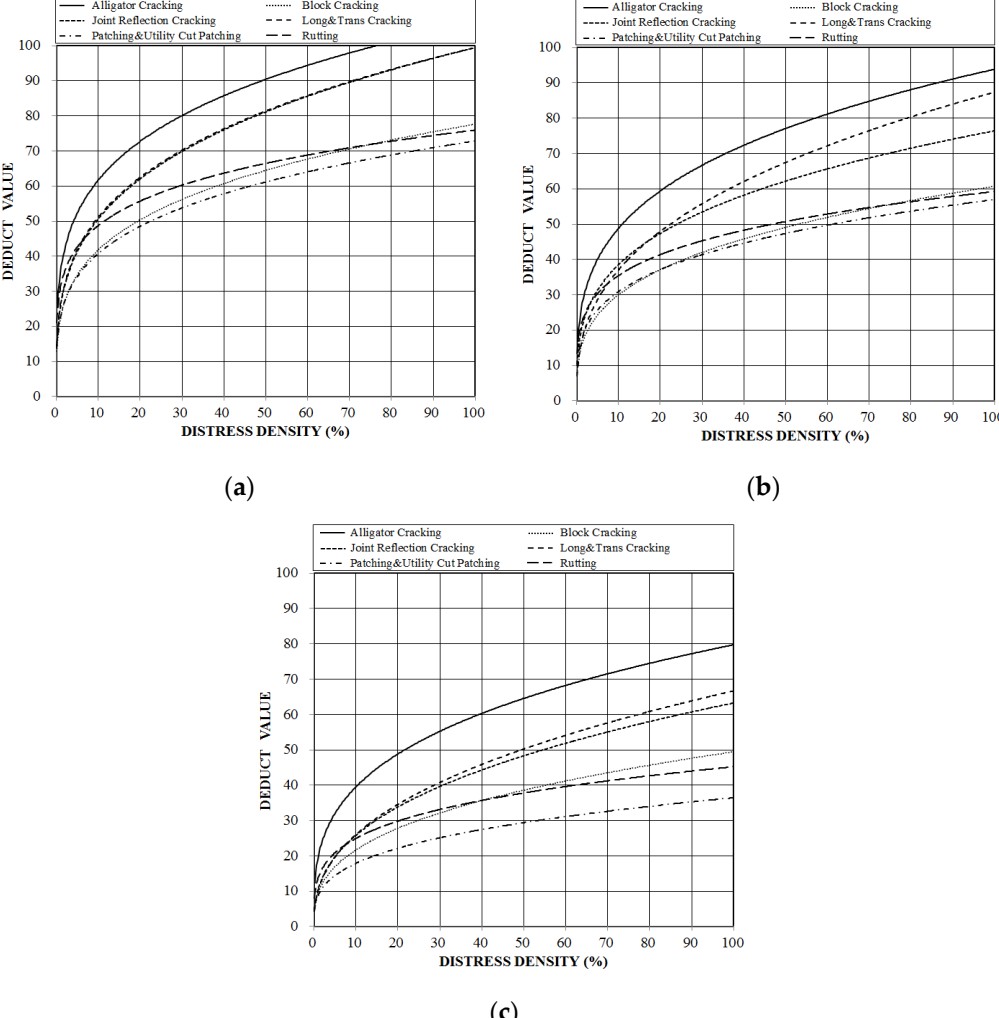

**Figure 6.** Developed deduct value curves of all distress types of asphalt pavement at (**a**) High Severity level, (**b**) Medium severity level, and (**c**) Low severity level.

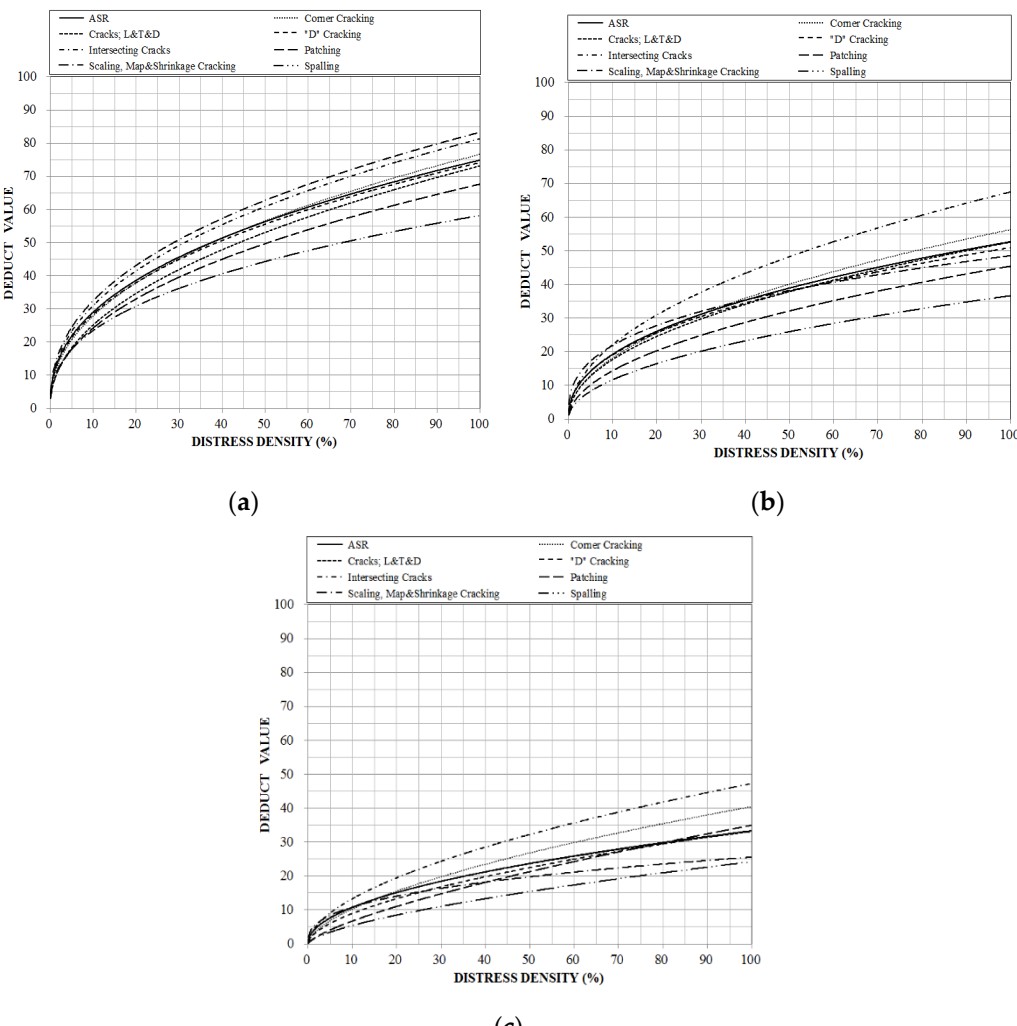

**Figure 7.** Developed deduct value curves of all distress types of concrete pavement at (**a**) high severity level, (**b**) medium severity level, and (**c**) low severity level.

Although most curves have similar shapes, their effects on the PCI may be very different. For example, the alligator cracking of the asphalt pavements has a significantly higher DV than the patching and intersecting cracks of the concrete pavements have a significantly higher DV than the spalling. Moreover, the curves change in different ways at each severity level. For example, as seen in the scaling, map cracks, and shrinkage cracks of the concrete pavements, the DV is low at the low severity level; however, the DV is the highest at the high severity level. Moreover, in case of the patching of the asphalt pavements, the DVC is relatively lower than other distresses, and it is found that an existing problem where the PCI becomes lower after the distress repair is largely solved.

Figure 8 shows the moving average CDV with respect to the TDV for a case with two or more distress types or severity levels in the asphalt pavement; and the CDVC, which is a regression curve passing through the average CDV. Figures 9 and 10 show the CDVC developed in this study for the asphalt and concrete pavements including two to five distress types or severity levels.

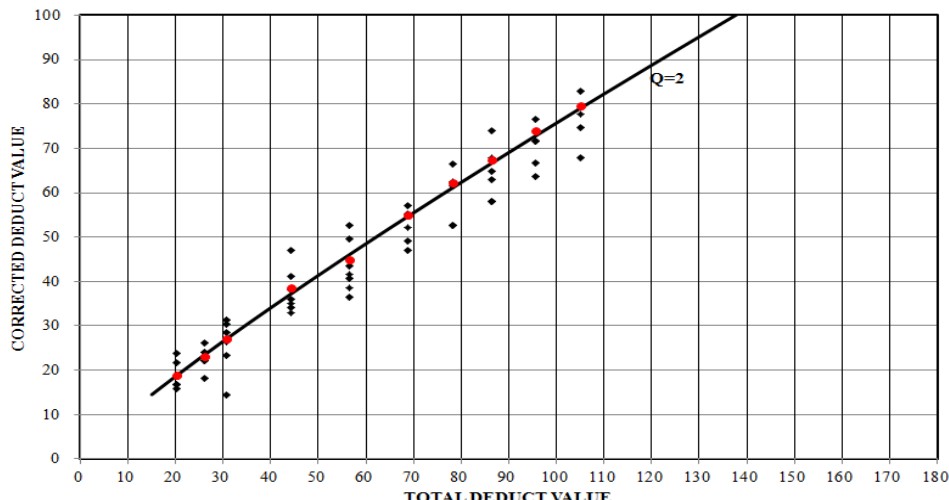

**Figure 8.** Example of the developed CDV for a case with two or more distress types or severity levels in the asphalt pavements.

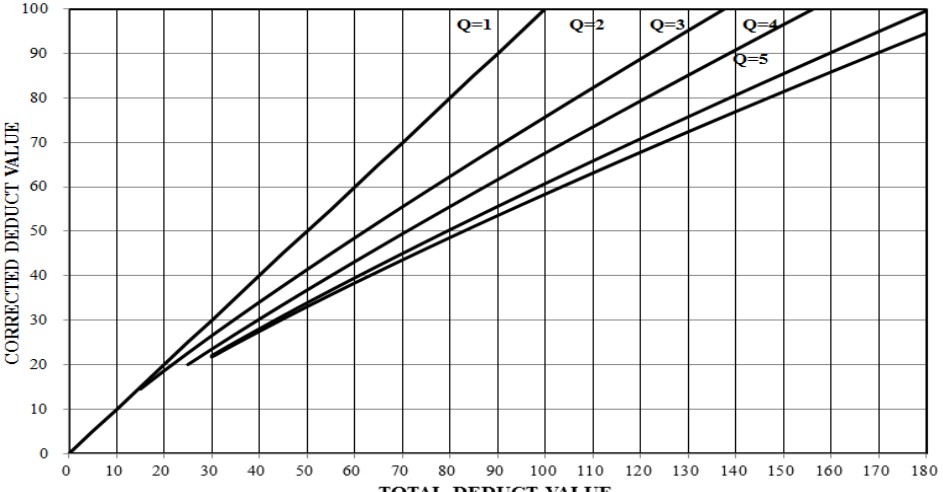

**Figure 9.** Developed corrected deduct value curves for asphalt pavement.

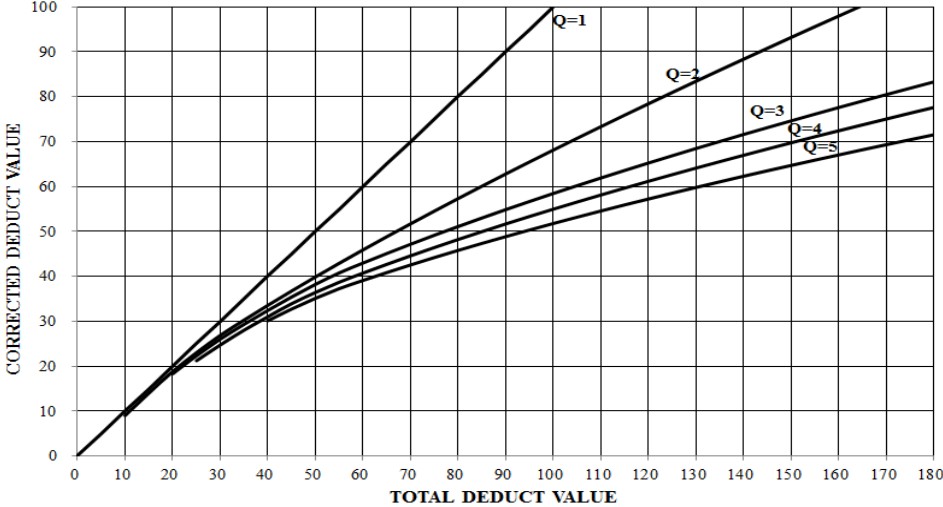

**Figure 10.** Developed corrected deduct value curves for concrete pavement.

*3.2. Evaluation and Field Validation*

3.2.1. Evaluation

The developed KPCI was compared and examined against the PCI calculated in accordance with ASTM D 5340 to evaluate its adequacy. To this end, 806 of asphalt pavements "sample unit" and 392 of concrete pavements "sample unit" of four Korean airports with secured real image data of the airport pavement surfaces were selected as targets. The PCI was derived using severity levels by distress of each criterion, deduct value curves, and corrected deduct value curves. There was no difference in the amount of distress even when the severity levels were different, as the distress determination is identical. Therefore, as the distribution of the amount of distress by severity in accordance with the criterion varies while the total amount of distress is constant and different DVCs are used, the PCI values also have differences. These results were compared to examine the overall adequacy of the developed criterion.

3.2.2. Comparison of PCI Calculation Results of ASTM D5340 and KPCI

Figure 10 shows the scattered plots for comparison between the criteria of PCI calculation results. The x-axis of the graph represents the PCI calculated using ASTM D 5340 and the y-axis represents the PCI calculated using the KPCI. The farther away from the diagonal line, the greater is the difference. The KPCI evaluates the pavement condition to be satisfactory in comparison to the PCI of ASTM D 5340 if a point is located in an area above the reference line, and the opposite is the case if it is in an area below the reference line. As a higher PCI means a more satisfactory pavement condition, as a result of comparison from such a viewpoint, the asphalt pavements, shown in Figure 11a, and the concrete pavements, shown in Figure 11b, are evaluated to be mostly lower than ASTM D5340 when evaluated by the KPCI. The asphalt pavements were evaluated to be 4.6 points lower on average, and the concrete pavements 6.9 points lower on average. The trend of being evaluated low became greater with worsening pavement condition and most sensitively decreased at approximately 70.

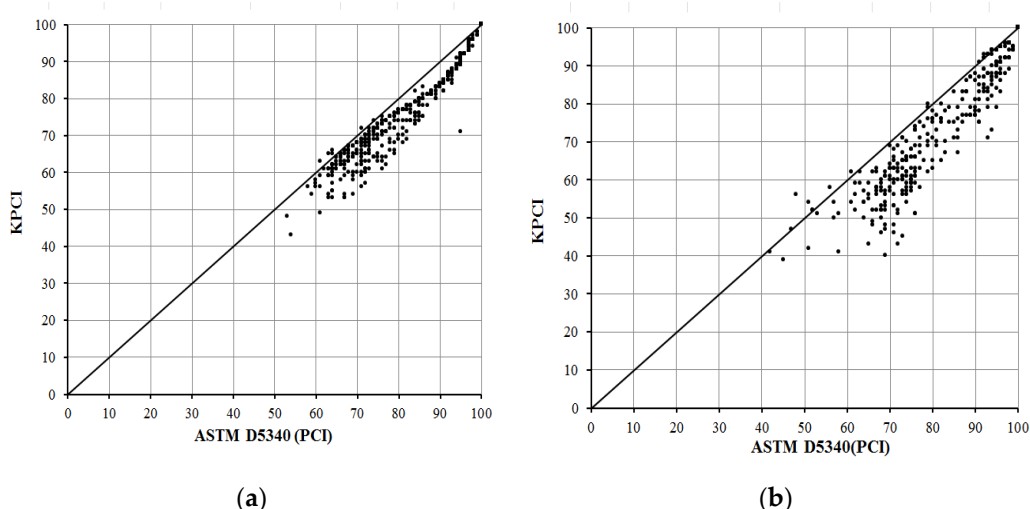

(**a**)                                        (**b**)

**Figure 11.** Comparisons of PCI between ASTM D 5340 and KPCI; (**a**) asphalt pavement, (**b**) concrete pavement.

**4. Findings**

In this paper, the Korea airport pavement condition index was developed by defining the main distresses of the Korean airports using an engineering-based approach, and deriving the deduct value curve of the main distresses observed in Korea. Following are the main research results of this paper:

1.  Six types (alligator cracking, block cracking, joint reflection cracking, longitudinal and transverse cracking, patching and utility cut patching, and rutting) of asphalt pavement distresses were classified as the main distresses of the Korean airports. Similar distresses among 16 types of concrete pavement distresses were combined to yield 13 types of distresses. Among them, eight types (ASR, corner cracking, cracks, durability cracking, intersecting cracks, patching, scaling, and spalling) were classified as the main distresses of the Korean airports.

2.  The DVC and CDV were derived by the pavement condition rating of the panel for the main distresses and multiple distresses for each pavement form. When the developed DVC and the DVC of ASTM D 5340 are compared, the DVCs were largely low or similar at the "High" and "Medium" severity levels and largely high at the "Low" severity level. The deduct value of the developed KPCI was higher as the amount of damage decreases in all three severity levels; however, it was lower than the existing PCI as the amount of damage increases. Moreover, the difference in the deduct values at each severity level was lower than the existing PCI. This is considered to be a result of incorporating the viewpoint of Korean airport pavement managers taking into consideration the importance of occurrences of distress even at low severity levels; and initial distress management.

3.  As a result of comparing the KPCI and PCI derived in accordance with ASTM D5340, the developed criteria of the asphalt pavements and concrete pavements showed an overall trend lower than that obtained by ASTM D5340. Moreover, it was verified that this trend increases with worsening pavement condition. It appears that a better discriminating evaluation may be possible when determining pavement conditions by PCI results of the developed criteria. As a result, the KPCI can be considered more suitable to derive evaluation results of a wider range than ASTM D5340. The pavement condition index should be capable of being used for distinguishing between good and defective and for reasonable determination of superiority and inferiority even at an equal level from the aspect of criteria required for maintenance and repair determination. The KPCI has this capability, which is favorable for application in Korea.

4.  It is considered that an existing problem where the PCI becomes lower after distress repair has been largely solved as the DVC is relatively lower than other distresses in the case of patching of asphalt pavements.

## 5. Conclusions

Considering that, in the Korean airports examined, most of the distress severity is distributed at low or medium levels, the effective causes of the difference in the PCI values of the samples analyzed for each pavement form are: (1) an increase in the severity level of distress by conservatively adjusting a portion of the determination criterion of the severity level by distress, and (2) the DV of the developed KPCI calculated to be higher as the amount of damage decreases in all three severity levels.

In comparison to the existing PCI, the KPCI appears to be capable of a better discriminating evaluation when determining pavement conditions.

**Author Contributions:** Conceptualization, N.-H.C. and Y.-C.S.; methodology, Y.-C.S.; validation, N.-H.C.; formal analysis, N.-H.C. and J.K.; investigation, J.K.; resources & data curation, H.-J.K.; writing—original draft preparation, N.-H.C.; writing—review and editing, J.K.; visualization, N.-H.C. All authors have read and agreed to the published version of the manuscript.

**Funding:** This research received no external funding.

**Informed Consent Statement:** Informed consent was obtained from all subjects involved in the study.

**Acknowledgments:** This paper was carried out by "Development of Construction and Maintenance Technology for Low-Carbon Green Airport Pavements" carried out by receiving support from the Korea Agency for Infrastructure Technology Advancement and research support of the Incheon International Airport Corporation. We are very grateful for their strong support.

**Conflicts of Interest:** The authors declare no conflict of interest.

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
