# Peer review of "Development of Korea Airport Pavement Condition Index for Panel Rating"

_applsci, doi:10.3390/app12168320_

Round 1
Reviewer 1 Report
This study examined the present PCI calculation process and then revised the major distress kind of Korean airport pavements and curves were used to develop the Korea airport pavement condition index (KPCI), and the field application results were compared to the current PCI to assess the KPCI's suitability. the paper is well-structured and written; however, there have been some comments should be considered:
Pls revise the abstract and include you major findings.
Pls revise the introduction and you are advised to refer more recent relevent studies and also authors should justify the problem statement more clearly.
Authors are encouraged to conduct more statistical analysis to show the correlation between the existing PCI and proposed KPCI method and also more in depth discussions are required for this section.
Author Response
Thank you for your careful review and valuable comments.
I revised the paper as attached by reflecting the opinion you gave me.
Thank you.

Reviewer 2 Report
This manuscript presents a study that developed the Pavement Condition Index (PCI) customized for Korea airport management. It provides a new example of PCI that is tailored to a specific application scenario. It is interesting to see how the PCI may vary from “standard” values when new dimensions of maintenance needs are incorporated in the development process of PCI.
The manuscript was generally well written with a clear description of the development process of the Korea PCI. The findings are supported by the analysis presented in the manuscript.
The manuscript may be considered for publication after the following minor comments are addressed.
1. Line 43 and Line 48, please revise the sentences to make them complete sentences. For example, who classified? Who considered?
2. Line 81, in “m2”, make “2” as a superscript.
3. Line 148, change “effected” to “affected”?
4. Line 182-184, please be more specific on how the CDV is calculated from the sum of DVs and a corrected value.
5. Line 209, change “;” to “,”
6. Line 280, revise “sIndeverity”.
7. In Figure 2, consider capitalizing the words in the axis labels so that they are consistent with other figures.
Author Response

(The authors gave the same response as above.)

Reviewer 3 Report
Dear Authors! The topic of the manuscript is very relevant! Therefore, I recommend finalizing the manuscript. In particular:
1) "... there have been difficulties in communication between maintenance and repair staff and decision makers" what difficulties?
2) «1. Introduction»
- "The index has provided sound engineering data for pavement conditions and has particularly been used for preventing potential problems, such as foreign object debris (FOD), and in decision-making processes for overall pavement maintenance and repair such as the prevention of additional damage due to cracks and the like." for clarity, I suggest adding photos of examples
- only [1-6] six literary sources?
I insist on rewriting "1. Introduction"!
3) "Figure 2. (a), (b)", and in the text simply "Figure 2"; "Figure 3. (a)-(f)", "Figure 4. (a)-(h)", and simply "Figures 3 and 4" in the text. And 5-6, 10. I propose to describe all the drawings in detail in the text
4) "References", the topic of the manuscript is very, very relevant and only 15 (???) literary sources! I recommend adding!!! Where is the research in the last 5 years?
Dear Authors, what is the "Originality", "Practical value" of your manuscript?
Dear Authors, my recommendation is "Reconsider after major revision".
Kind regards, L. Neduzha
Author Response

(The authors gave the same response as above.)

Reviewer 4 Report
1. The authors were recommended to revise the abstract section to better highlight methodology used in the study. Besides, some research findings were suggested to add in the abstract section as well.
2. Please provide more detailed explanations about the potential problems and limitations concerning the statement “However, potential problems and limitations pointed out when it comes to the applicability of the PCI calculation procedure in accordance with the differences between the environment at the time of development and the current environment. ”
3. I want to know whether there was an operator before the F(t, d) in Eq. (1)?
4. How did authors select comparison target to existing PCI in Figure 1? Please provide more explanations.
5. How did the authors obtain formula y=8.6736x0.463 in figure 2(b)? Another problem was the formula successfully fit with the data distributions?
6. The following studies were recommended to be properly cited: [1]X. Chen, S. Wu, C. Shi, Y. Huang, Y. Yang, R. Ke, et al., Sensing Data Supported Traffic Flow Prediction via Denoising Schemes and ANN: A Comparison," IEEE Sensors Journal, vol. 20, pp. 14317-14328, 2020. [2] K. P. Ziock et al., "Performance of the Roadside Tracker Portal-Less Portal Monitor," in IEEE Transactions on Nuclear Science, vol. 60, no. 3, pp. 2237-2246, June 2013, doi: 10.1109/TNS.2013.2262472.
Author Response

(The authors gave the same response as above.)

Round 2
Reviewer 3 Report
Dear authors, thank you for your work
regards,
Neduzha Larysa
Author Response
I revised the paper by broadcasting the main opinion as attached.
Thank you for your opinion.

Reviewer 4 Report
My comments have been addressed.
Author Response
Thank you for your excellent review.